# MoA: Mixture of Aggregators Improves Slide-Level Diagnosis in Computational Pathology

**Fatih Ozlugedik**[*1]                                   FATIH.OEZLUEGEDIK@HELMHOLTZ-MUNICH.DE

[1] *Institute of AI for Health, Helmholtz Munich, Munich, Germany*

**Muhammed Furkan Dasdelen**[*1] iD            FURKAN.DASDELEN@HELMHOLTZ-MUNICH.DE

**Rao Muhammad Umer**[1] iD                          UMER.RAO@HELMHOLTZ-MUNICH.DE

**Carsten Marr**[1,2,3,4] iD                              CARSTEN.MARR@HELMHOLTZ-MUNICH.DE

[2] *Department of Medicine III, Ludwig-Maximilian-University Hospital, Munich, Germany*

[3] *Munich Center for Machine Learning (MCML), Munich, Germany*

[4] *DKTK, German Cancer Consortium, Munich, Germany*

**Editors:** Accepted for publication at MIDL 2026

## Abstract

Multiple instance learning (MIL) is the standard for learning slide-level representations from whole slide images (WSIs), typically using a single attention-based aggregator to pool instance features. However, a single aggregator can struggle to capture morphological and compositional patterns of cells in pathology and cytology data, and different diseases may demand different pooling behaviours. We propose a mixture-of-aggregators framework that models complementary aspects of instance distributions in histology and hematologic cytology. A router with top-2 gating dynamically selects the most relevant aggregators per slide, and their outputs are fused into a patient-level representation. To avoid collapse to a single dominant expert aggregator, we add a load-balancing loss and Gumbel noise on the router logits. We evaluate our method on 19 different tasks from 16 datasets including histology and hematologic cytology. Compared to single-aggregator baselines, our approach improves diagnostic prediction accuracy by an average of 4.5% over ABMIL and 12.6% over TransMIL across all tasks. Beyond performance, our analysis shows that different aggregators attend to distinct, disease-specific instance distributions, providing interpretable insights into the diagnostic process.

**Keywords:** computational pathology, multiple instance learning, cytology

## 1. Introduction

Pathology whole-slide images (WSIs) are indispensable for cancer diagnosis, but their manual assessment is time-consuming and highly dependent on expert experience. With the advent of high-throughput slide digitization, AI-based approaches have been introduced to support tumor detection, grading, morphological and molecular subtyping, and even survival prediction (Chen et al., 2024; Lu et al., 2023; Bilal et al., 2021; Vorontsov et al., 2024; Li et al., 2023). These tools can reduce the workload of pathologists while enabling faster and more standardized diagnostic outputs (Bulten et al., 2021; Dy et al., 2024; Steiner et al., 2018; Janowczyk et al., 2019). WSIs are extremely large, often reaching gigapixel resolution, yet typically come with only slide-level labels. To bridge this gap, multiple instance learning (MIL) has become the standard paradigm. In MIL, a WSI is partitioned

---

[*] Contributed equally

into non-overlapping patches, each patch is encoded into a feature representation, and an aggregator combines these patch-level features into a slide-level embedding for downstream tasks (Li et al., 2021; Lu et al., 2021; Campanella et al., 2019; Ilse et al., 2018; Shao et al., 2021; Ding et al., 2024).

The success of MIL depends critically on both the quality of instance encoding and the design of the aggregator. Modern pathology foundation models provide strong instance-level features (Filiot et al., 2024; Bioptimus, 2025; Zimmermann et al., 2024; Chen et al., 2024; Lu et al., 2024), but aggregation remains challenging (Chen et al., 2024; Ding et al., 2024). Most current MIL methods focus on identifying a small subset of diagnostically relevant patches, which is effective when the presence of a single pattern is sufficient for diagnosis. However, many diseases are defined not only by the presence of specific cell types or structures, but also by their distribution and relative frequency within the slide. Conventional single-aggregator approaches are prone to fail to capture these subtler distributional patterns, leading to a loss of diagnostically important information (Lu et al., 2021; Li et al., 2021; Shao et al., 2021).

Mixture-of-experts strategies, widely adopted in large language models, show that dividing responsibility across multiple specialized components allows the system to model diverse tasks and distributions more effectively (Jacobs et al., 1991; Shazeer et al., 2017; Riquelme et al., 2021). Inspired by this idea, we propose a mixture of aggregators for computational pathology. Within a single pipeline, multiple aggregators can learn complementary aspects of slide composition—some focusing on highly discriminative instances, others capturing broader distributional signals. We hypothesize that such diversity enables the model to represent distinct disease-specific distributions more faithfully, leading to improved diagnostic performance and better alignment with clinical reasoning. The main contributions of our work are: (i) Instead of a single aggregator, we train multiple aggregators in a MIL pipeline using a routing strategy that weights each aggregator's contribution. (ii) Our pipeline supports diverse aggregator architectures and improves performance. (iii) We show that each aggregator captures distinct, diagnostically relevant, and complementary instance distributions.

## 2. Related work

### 2.1. Aggregators in multiple instance learning

Early multiple instance learning (MIL) approaches for whole-slide images (WSIs) employed non-parametric, permutation-invariant pooling functions—such as mean, max, and log-sum-exp (LSE)—to compress instance features into slide-level representations (Campanella et al., 2019; Ilse et al., 2020; Keshvarikhojasteh, 2025). While simple and efficient, these fixed functions have limited capacity to adapt to data.

A major advance beyond static pooling mechanisms was the introduction of Attention-based Multiple Instance Learning (ABMIL) (Ilse et al., 2018; Sadafi et al., 2020). AB-MIL learns instance-specific attention weights and computes a weighted average of patch embeddings, enabling more flexible slide-level predictions and interpretable instance-level heatmaps. Building on ABMIL, several extensions have been proposed. Clustering-constrained Attention MIL (CLAM) incorporates instance-level clustering to promote diverse class-specific prototypes (Lu et al., 2021), while Dual-stream MIL (DSMIL) couples an instance-

discriminative stream with a bag-level stream through contrastive alignment (Li et al., 2021). More recently, Transformer-based aggregators such as TransMIL apply self-attention across patches, explicitly modeling inter-instance relationships and often achieving improved whole-slide accuracy (Shao et al., 2021).

Despite their architectural differences, these MIL approaches share a common bottleneck in how they form the final slide-level representation. In most implementations—including ABMIL, DSMIL, and Transformer-based MIL models—the bag is ultimately reduced to a pooling operation. This is typically realized either through attention pooling or through a classification token (CLS) whose final hidden state $h_{\text{cls}}^{(L)}$ summarizes the entire set. Such readouts are mathematically equivalent to Pooling by Multi-Head Attention (PMA) with $k = 1$ in Set Transformers (Lee et al., 2019), and fall under the Deep Sets formulation (Zaheer et al., 2017). In essence, the model relies on a single learned query vector that attends over all patches, producing a learned weighted first-order moment of the instance distribution.

While this mechanism effectively captures average signal, it cannot directly model higher-order statistics—such as co-occurrence structures, multimodal feature distributions, or rare-pattern enrichment—which are central to histological heterogeneity and diagnostic accuracy. As a consequence, the architecture implicitly assumes that a slide can be summarized by a single global prototype. This assumption routinely breaks down in heterogeneous whole-slide images, where multiple competing morphologies or subclonal populations may coexist (Zaheer et al., 2017; Lee et al., 2019; Dosovitskiy et al., 2021; Shao et al., 2021).

CLAM partially addresses this issue by using multi-head attention designed to produce class-specific attention maps, but all heads still share a common backbone, limiting their representational diversity. Transformer-based MIL methods in principle can capture richer distributions via self-attention, yet the quadratic complexity of standard attention becomes prohibitive for thousands of patches. TransMIL mitigates this through hierarchical processing with neighbor-restricted (windowed) attentions and cross-scale fusion, reducing effective complexity while retaining contextual information. However, its hierarchical design introduces permutation variance and reduces interpretability—limitations that are problematic for inherently permutation-invariant domains such as cytology.

These observations collectively suggest that effective slide-level analysis requires multiple specialized aggregation mechanisms that can adapt to different morphological patterns within a slide. This naturally points toward architectures that dynamically select or combine diverse processors rather than relying on a single monolithic pooling mechanism.

## 2.2. Mixture-of-Experts for specialized modeling

The Mixture-of-Experts (MoE) framework is a well-established paradigm for scaling model capacity efficiently by employing a set of specialized sub-networks ("experts") and a learned router that allocates inputs to the most relevant experts. Each expert has its own weight space, enabling specialization without forcing the model to optimize a single compromised solution across partially contradicting objectives (Jacobs et al., 1991; Shazeer et al., 2017). While MoE has been widely adopted in natural language processing, its application in computational pathology has been limited, particularly at the critical aggregation stage.

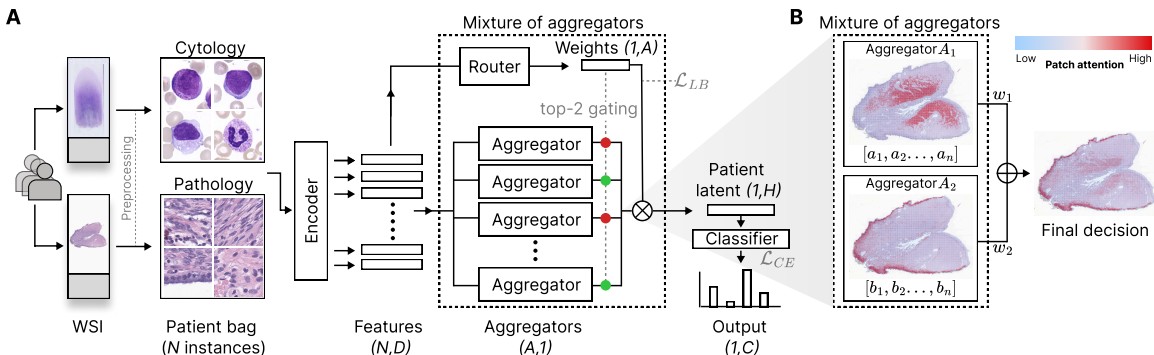

Figure 1: (A) Whole-slide images are patched and encoded into instance-level embeddings. A router assigns weights and selects the top-2 aggregators, which process the embeddings in parallel. Their weighted outputs are fused into a patient-level latent representation and passed to a classifier for disease prediction. (B) Each aggregator learns to focus on distinct morphological structures within the slide.

Prior work has explored MoE in computational pathology primarily in multi-task learning settings, where experts are used to share representations across related tasks (e.g., mutation prediction) via task-specific multi-gated routing (Li et al., 2025). In this study, we introduce a slide-level MoE approach that combines multiple aggregators under a single router, which we term Mixture of Aggregators (MoA). In contrast to (Li et al., 2025), our work focuses on single-task diagnosis and aims to capture slide-level heterogeneity.

Instead of a single aggregator, several permutation-invariant aggregators are trained in parallel, each free to specialize in distinct morphological or domain regimes (for example, immune-rich versus tumor-dominant patterns, rare event sensitivity, or co-occurrence structures). A router produces data-dependent weights (often sparse top-$k$) that combine the aggregator summaries for each slide. This preserves a shared feature backbone while adding specialized capacity exactly where heterogeneity is highest—during the instance-to-slide aggregation. In effect, combining multiple invariant summaries via routing enables the model to approximate a richer family of set functions than any single aggregator, while keeping the interface simple (a slide-level vector) and the training recipe close to standard MIL practice (Zaheer et al., 2017; Lee et al., 2019; Shazeer et al., 2017; Lepikhin et al., 2020).

## 3. Methodology

### 3.1. MoA: Mixture of aggregators in multiple instance learning

Our framework employs multiple aggregators to capture distinct distributions of instances across different disease types. In our mixture of aggregator experiments, we use two commonly adopted aggregator architectures: Attention-based MIL (ABMIL) and a Transformer-based aggregator. Although many variants of aggregators exist and in theory one may appear superior to another, recent studies show that their performance is often data-specific

and strongly influenced by hyperparameter tuning (Shao et al., 2025). In comparison, we include CLAM, DSMIL and mean pooling (MeanMIL).

For ABMIL, we use the gated attention pooling mechanism of Ilse et al. (Ilse et al., 2018), which learns instance-specific attention weights based on nonlinear gating. The bag representation is then obtained as a weighted sum of the instance embeddings.

As Transformer-based aggregators, we use permutation invariant Transformer proposed by Wagner et al (Wagner et al., 2023) for cytology tasks. Each instance embedding is projected into a 512-dimensional latent space, and a learnable [CLS] token is prepended to the sequence. We use two transformer layers, each consisting of a multi-head self-attention block (with 8 heads) followed by a feedforward network with hidden dimension 1024. For pathology tasks, we use TransMIL (Shao et al., 2021) with same dimensional parameters. [CLS] token is later used as the aggregated representation.

## 3.2. Architecture

We first use an encoder to extract instance-level features (Figure 1). For pathology images, we use UNI (Chen et al., 2024), and for hematologic cytology we use DinoBloom-B (Koch et al., 2024). The extracted features of size (N,D) are passed through a router to compute aggregator weights. The router consists of a projection layer, feature mean pooling and a linear layer. A top-2 softmax gating then selects the most relevant aggregators.

In parallel, the features are passed through all aggregators (Figure 1). The representations from the top-2 selected aggregators are weighted and summed to produce the patient-level latent representation, which is subsequently used for classification.

For optimization, we use standard cross-entropy loss ($\mathcal{L}_{\mathrm{CE}}$) for slide-level classification and add a load-balancing auxiliary loss on the gating network to avoid routing collapse. This loss, following (Fedus et al., 2021), encourages all aggregators to be used more evenly. Let $A$ be the number of aggregators and $\mathcal{B}$ the effective batch of $T$ patient-level bags. During training, we use top-2 routing, i.e., each bag is sent to the two aggregators with the highest router probabilities. For each aggregator $i \in \{1, \ldots, A\}$ we define

$$f_i = \frac{1}{T} \sum_{x \in \mathcal{B}} \mathbf{1}\left[i \in \mathrm{TopK}(p(x), 2)\right], \quad P_i = \frac{1}{T} \sum_{x \in \mathcal{B}} p_i(x) \tag{1}$$

where $p(x) \in \mathbb{R}^A$ is the router's softmax probability vector for patient $x$, $p_i(x)$ its $i$-th component, and $\mathrm{TopK}(p(x), 2)$ returns the indices of the two aggregators with highest probability. Thus, $f_i$ is the fraction of bags actually routed to aggregator $i$ (hard usage), while $P_i$ is the average router probability mass assigned to aggregator $i$ (soft usage). The load-balancing loss is

$$\mathcal{L}_{\mathrm{LB}} = A \sum_{i=1}^{A} f_i P_i, \quad \mathcal{L} = \mathcal{L}_{\mathrm{CE}} + \lambda_{\mathrm{lb}} \cdot \mathcal{L}_{\mathrm{LB}}, \tag{2}$$

where $\lambda_{\mathrm{lb}}$ controls the strength of load balancing.

In addition, to mitigate aggregator collapse and promote diverse aggregator utilization, we perturb the gating logits with independent Gumbel(0,1) noise during training (Shazeer et al., 2017). To gradually transition from exploration to stable specialization, we anneal

the softmax temperature: higher temperatures produce smoother, more exploratory distributions across aggregators, while lower temperatures sharpen the distribution and favor more deterministic routing (Nie et al., 2022).

## 4. Experiments

### 4.1. Dataset and preprocessing

We test our model on 2 modalities (cytology and histology), 13 organs/regions and 19 tasks including morphological and immune subtyping. All datasets are publicly available. More details are included in Appendix A.3.

**Cytology**: AML-Hehr (Hehr et al., 2023) and cAItomorph (Dasdelen et al., 2026) are blood smear datasets which include single cell white blood cell images.

**Pathology**: We include fourteen different pathology datasets in our evaluation. These span multiple organ systems, including breast pathology (BCNB (Xu et al., 2021), BRACS (Brancati et al., 2022)), renal cancer (CPTAC-CCRCC), head and neck cancer (CPTAC-HNSC, HANCOCK (Dörrich et al., 2025)), lung cancer (CPTAC-LSCC), pancreatic cancer (CPTAC-PDA), endometrial cancer (CPTAC-UCEC), cervical cancer (IMP-Cervix (Oliveira et al., 2024)) and other (CPTAC-ALL). ((Ellis et al., 2013; Zhang et al., 2025))). Tasks include biomarker prediction (BCNB), immune class and tumor microenvironment prediction (CPTAC-CCRC/HNSC/PDA/UCEC, HANCOCK), histological grading (CPTAC/LSCC, HANCOCK, IMP-CERVIX) and tumor site prediction (CPTAC-ALL, HANCOCK).

For all datasets, we fix the test set according to the original publications or the benchmark (Zhang et al., 2025; Vaidya et al., 2025) and report model performances on test set. Within the training split, we perform 5-fold cross-validation.

For cytology datasets, we use the DinoBloom-B hematology feature extractor (Koch et al., 2024). We patchify pathology datasets, using TRIDENT (Zhang et al., 2025; Vaidya et al., 2025) pipeline at $20\times$ magnification, patch size of $256 \times 256$. These patches are then embedded using the UNI feature extractor (Chen et al., 2024).

### 4.2. Evaluation metrics

For multi-class tasks, we report balanced accuracy, while for binary tasks, we report the area under the ROC curve (AUROC).

To analyze instance-level attentions, we extract the attention scores produced by each aggregator. For ABMIL, we directly use the learned attention weights assigned to individual instances. For Transformer-based aggregators, we apply the Attention Rollout method (Abnar and Zuidema, 2020).

Jensen–Shannon divergence (JSD) is utilized to quantify attention distribution differences of aggregators (Lin, 2002).

Table 1: Comparison of aggregators across datasets. Δ indicates relative improvement compared to single aggregator (%). The reported metric is balanced accuracy for multi-class, area under the ROC (AUROC) for binary tasks. **Bold** indicates the best performing model between single vs. mixture of aggregators for the same architecture. Underline highlights the best model across all models.

| Dataset (Number of class) | MoA-ABMIL | ABMIL | Δ | MoA-TransMIL | TransMIL | Δ | CLAM-SB | DSMIL | MeanMIL |
|---|---|---|---|---|---|---|---|---|---|
| AML-Hehr (5C) | 78.4±2.2 | **81.5±3.7** | -3.8% | **81.5±1.0** | 78.6±2.2 | +3.7% | 76.7±6.0 | 41.7±2.7 | 78.3±2.1 |
| cAItomorph (8C) | **52.8±1.8** | 50.9±1.9 | +3.7% | **60.1±1.1** | 59.4±1.9 | +1.2% | 55.0±3.0 | 42.0±2.5 | 60.1±1.4 |
| BCNB/ER (2C) | **91.3±0.2** | 91.2±0.4 | +0.1% | **88.4±0.5** | 85.7±3.3 | +3.1% | 90.8±0.7 | 90.9±0.5 | 91.7±0.4 |
| BCNB/HER2 (2C) | **84.0±0.7** | 84.0±0.5 | 0.0% | **81.2±2.2** | 69.8±3.5 | +16.4% | 83.1±2.2 | 81.5±1.3 | 84.1±1.1 |
| BCNB/PR (2C) | **88.3±0.4** | 88.2±0.6 | +0.1% | **84.7±0.7** | 78.2±3.5 | +8.3% | 87.1±1.2 | 86.4±0.4 | 89.0±0.5 |
| BRACS (7C) | **34.6±1.9** | 34.4±1.5 | +0.6% | **29.3±1.6** | 26.5±1.5 | +10.6% | 32.8±3.3 | 27.2±2.5 | 26.9±1.7 |
| CPTAC-ALL (10C) | **96.1±0.3** | 95.5±0.2 | +0.6% | **96.3±0.6** | 95.5±1.0 | +0.8% | 96.4±0.7 | 96.5±0.4 | 96.8±0.6 |
| CPTAC-CCRCC (3C) | **45.4±7.7** | 43.5±4.3 | +4.4% | **47.2±4.1** | 45.2±3.7 | +4.4% | 47.7±4.0 | 45.4±3.7 | 45.9±4.7 |
| CPTAC-HNSC (3C) | **35.1±5.7** | 33.0±4.6 | +6.4% | **31.6±3.6** | 27.9±5.8 | +13.3% | 35.1±3.9 | 30.2±3.0 | 34.5±5.6 |
| CPTAC-LSCC (2C) | **69.8±2.2** | 67.1±3.5 | +4.0% | **63.7±5.2** | 60.0±9.4 | +6.2% | 65.1±2.3 | 65.0±2.8 | 60.2±3.4 |
| CPTAC-PDA (3C) | **39.3±7.3** | 35.1±3.0 | +11.9% | **41.3±4.1** | 32.9±7.0 | +25.5% | 40.0±6.1 | 36.2±3.1 | 40.1±2.6 |
| CPTAC-UCEC (3C) | **43.2±2.8** | 36.3±5.6 | +19.0% | **44.9±7.3** | 29.7±7.9 | +51.2% | 37.0±9.4 | 28.7±7.8 | 33.5±10.1 |
| HANCOCK/K-SCC grading (2C) | **73.8±1.3** | 71.4±5.8 | +3.4% | **73.6±2.9** | 60.8±6.6 | +21.1% | 67.6±2.3 | 54.7±9.2 | 70.8±4.9 |
| HANCOCK/NK-SCC grading (2C) | **67.0±5.8** | 62.0±10.8 | +8.1% | **61.0±10.2** | 48.0±9.7 | +27.1% | 62.5±8.2 | 46.0±15.5 | 53.0±12.2 |
| HANCOCK/perineural invasion (2C) | **79.8±1.1** | 76.9±0.7 | +3.8% | **75.5±3.2** | 63.9±5.6 | +18.1% | 75.6±2.9 | 63.0±7.8 | 76.1±0.7 |
| HANCOCK/metastasis (2C) | **74.8±1.3** | 71.4±1.7 | +4.8% | **64.7±3.6** | 63.2±5.5 | +2.4% | 67.1±6.5 | 62.6±6.8 | 73.3±3.6 |
| HANCOCK/tumor site (4C) | **74.1±3.2** | 68.5±1.9 | +8.2% | **71.8±3.2** | 66.7±1.0 | +7.6% | 71.3±2.0 | 60.4±2.2 | 71.1±2.4 |
| HANCOCK/vascular invasion (2C) | **55.3±6.8** | 51.6±7.6 | +7.2% | **66.8±3.8** | 59.9±6.0 | +11.5% | 62.1±6.4 | 52.8±8.0 | 55.3±9.9 |
| IMP-Cervix (3C) | **46.6±2.8** | 45.0±3.6 | +3.6% | **57.0±1.9** | 52.9±4.5 | +7.7% | 61.0±4.6 | 47.0±6.7 | 48.9±1.9 |
| **Average** | **64.7** | 62.5 | **+4.5%** | **64.2** | 58.1 | **+12.6%** | 63.9 | 55.7 | 62.6 |

## 4.3. Results

### 4.3.1. MoA enhances diagnostic predictions

We evaluate MoA across two modalities (cytology and pathology) and 19 downstream tasks (Table 1). A single, fixed training recipe—selected via the AML-Hehr ablations—is applied consistently to all datasets.

For ABMIL experts, MoA-ABMIL matches or exceeds the single-aggregator ABMIL baseline in almost all settings. Across 19 tasks, MoA-ABMIL underperforms the baseline only once, while providing consistent positive or neutral gains elsewhere. Improvements are marginal on high-performing binary pathology tasks (e.g., BCNB, CPTAC-ALL; Δ ≈ 0.1−0.8%), and become more pronounced on harder multi-class cohorts with lower baselines. For example, MoA-ABMIL improves balanced accuracy on CPTAC-HNSC (+6.4%), CPTAC-LSCC (+4.0%), CPTAC-PDA (+11.9%) and CPTAC-UCEC (+19.0%), and achieves the best overall performance on several HANCOCK grading and invasion tasks. On average, MoA-ABMIL yields a +4.5% relative improvement over the single-aggregator ABMIL baseline across all datasets.

For mixture of Transformer experts, the effect is even stronger. MoA-TransMIL strictly dominates the single TransMIL baseline on every dataset, with an average relative gain of +12.6% (Table 1). Improvements are modest on already saturated tasks (e.g., CPTAC-ALL: +0.8%), but become substantial on more challenging settings: BRACS (+10.6%), BCNB/HER2 (+16.4%), BCNB/PR (+8.3%), CPTAC-PDA (+25.5%), and especially CPTAC-UCEC (+51.2%). Similar trends are observed across the HANCOCK, where MoA-TransMIL delivers large gains for K-SCC grading (+21.1%), NK-SCC grading (+27.1%), perineural invasion (+18.1%), vascular invasion (+11.5%). In cytology, MoA-TransMIL also

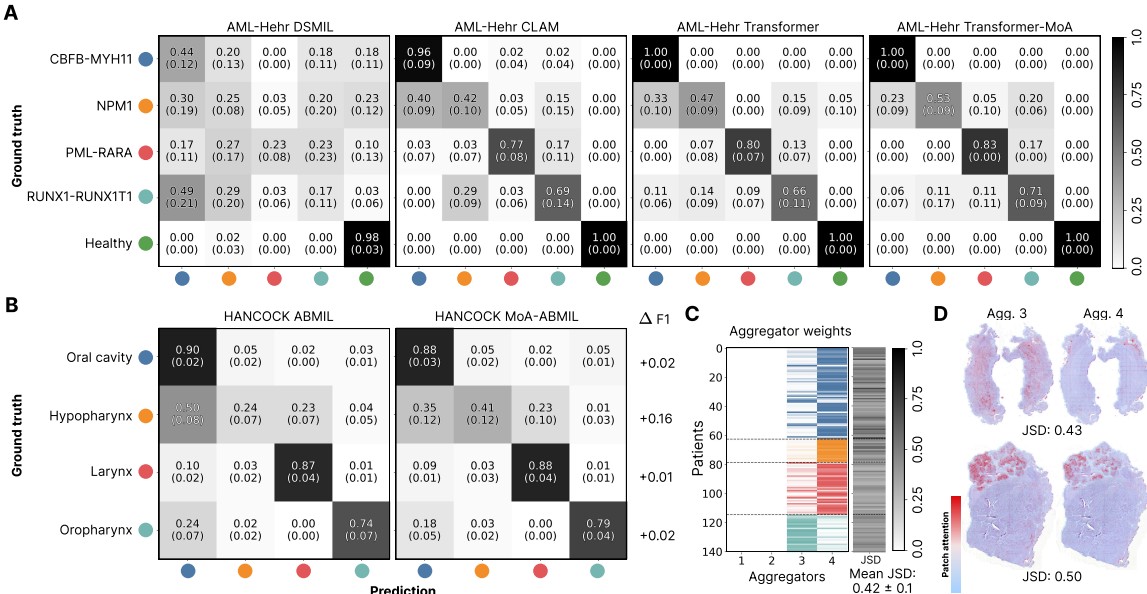

Figure 2: (A) Confusion matrices for AML-Hehr dataset. (B) MoA improves class specific F1 scores. (C) Aggregator weight distributions across patients reveal class-specific specialization. (D) Attentions given by aggregators differ for patients and quantified by Jensen–Shannon divergence (JSD).

improves over the baseline for AML-Hehr (+4.5%) and cAItomorph (+1.2%), confirming that the benefits of mixture-of-aggregators transfer to both smear and tissue-based tasks.

Compared to alternative MIL baselines, MoA is competitive or superior on most datasets. Although the simplest, MeanMIL attains the best performance on 4/19 tasks (BCNB and CPTC-ALL), while CLAM-SB slightly outperforms other models on CPTAC-CCRCC and IMP-Cervix. Nevertheless, either MoA-ABMIL or MoA-TransMIL is the best or tied-best model on the majority of tasks (13/19). These results indicate that mixing aggregators–regardless of the choice of backbone–yields a robust improvement over both single-aggregator variants and established MIL baselines.

The mixture of ABMIL and TransMIL aggregators within the MoA framework (2× ABMIL + 2× TransMIL) does not provide additional benefit and often performs between the pure ABMIL MoA and pure TransMIL MoA (Appendix Table 3).

The additional benefit MoA on cytology and histology tasks are shown in Figure 2 with confusion matrices. Beyond overall performance, our method enhances class-level sensitivity and reduces confusion between malignant and non-malignant categories in the AML-Hehr dataset (Figure 2A). Our method consistently improve performance of all classes in HAN-COCK dataset (Figure 2B). Importantly, the aggregator weight distributions reveal distinct specializations (Figure 2C). In the primary tumor-site identification task, Aggregator 3 predominantly contributes to oropharynx cases, while Aggregator 4 is more active in other tumor regions. Jensen–Shannon divergence analysis shows that the aggregators attain different attention distributions over patches (Figure 2C, D), with a mean of $0.42 \pm 0.10$.

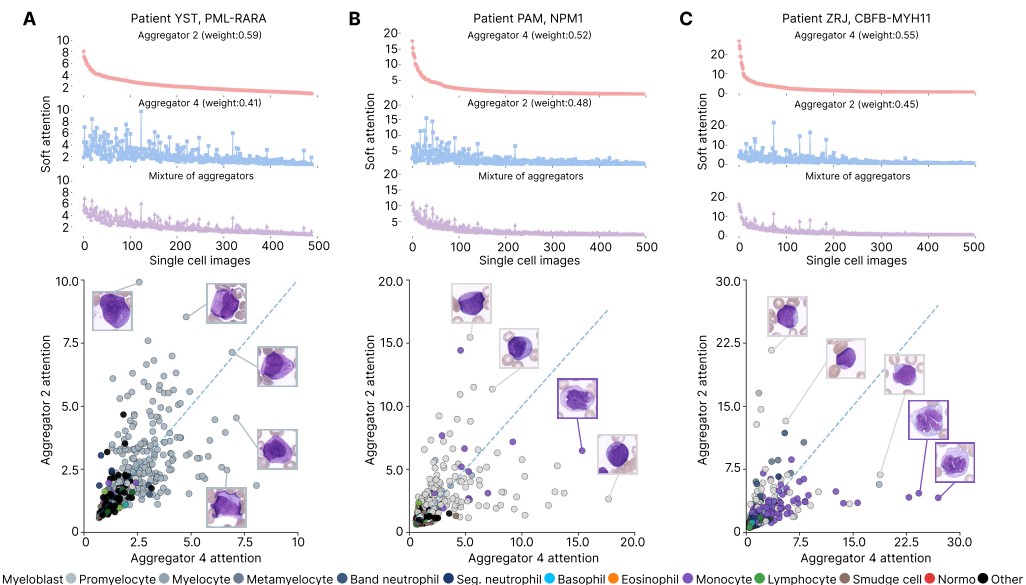

Figure 3: Different aggregators capture complementary diagnostic patterns. For the AML-Hehr dataset, we show patient-level attention analyses for (A) AML with PML-RARA, (B) AML with NPM1, and (C) AML with CBFB-MYH11. Top panels display single-cell attention distributions for the top-2 contributing aggregators and their mixture weights (red: dominant, blue: secondary, purple: mixture). Bottom panels compare their attentions, illustrating how each focuses on distinct cell subsets. This complementarity enables the model to capture subtype-specific morphological patterns. Attention scores are reported in $10^{-3}$ units.

### 4.3.2. Aggregators capture distinct attention patterns

To further assess aggregator specialization and determine whether they capture different distributions within a bag, we conducted a patient-wise analysis and computed instance-level attentions (single-cell images) from each aggregator (Figure 3). We focus on the AML-Hehr hematologic cytology dataset because single-cell contributions to disease are easier to assess and several AML subtypes exhibit pathognomonic morphologic findings. We present three representative patients from different AML subtypes. The upper panel shows the attention distributions generated by the aggregators along with their respective contribution weights.

Patient YST carries a PML::RARA fusion, also known as acute promyelocytic leukemia (APL), a distinct subtype of myeloid leukemia that requires rapid diagnosis and treatment. The hallmark of APL is the presence of promyelocytes. In patient YST, the dominant aggregator (Aggregator 2) assigns high attention to several promyelocytes, while the second aggregator highlights additional promyelocytes initially overlooked by the first (Figure 3A).

In patient PAM, the aggregators complement each other, with both capturing myeloblasts, which are essential for AML diagnosis (Figure 3B).

Patient ZRJ harbors a CBFB::MYH11 fusion, a genetic abnormality associated with AML characterized by monocytic and granulocytic differentiation. In this case, the first ag-

gregator (Aggregator 4) identifies monocytic cells as disease-specific instances (Figure 3C). The second aggregator complements by assigning high attention to myeloblasts and granulocytic cells at different maturation stages.

For quantitative evaluation, we calculated the intersection over union (IoU) between the high-attention top-k fraction of cells selected by different aggregators (Appendix Figure 4). The mean IoU across test samples is $0.16 \pm 0.17$ at $k = 0.01$ and $0.21 \pm 0.10$ at $k = 0.05$, showing that only a small fraction of highly attended cells are shared between aggregators.

## 4.4. Ablation Study

Table 2: Selected router configurations on AML-Hehr. *Baseline* = single aggregator, no router; *MLP* = multi-layer perceptron; *# aggregators* = number of aggregators; $\lambda_{\mathrm{lb}}$ = load-balancing loss coefficient. Use of Gumbel routing is indicated as True (T)/False (F). **Bold** indicates the best-performing configuration.

| Hyperparameter | Router arch | # aggregators | Top-k | $\lambda_{\mathrm{lb}}$ | Gumbel | bAcc ($\Delta$) |
|---|---|---|---|---|---|---|
| Baseline | - | - | - | - | - | 78.6 |
| Default | Linear | 4 | 2 | 0.01 | T | **81.5**(**+3.7**) |
| Router arch | MLP | 4 | 2 | 0.01 | T | 76.1($-3.2$) |
| # aggregators | Linear | 2 | 2 | 0.01 | T | 76.8($-2.3$) |
| # aggregators | Linear | 6 | 2 | 0.01 | T | 79.3($+0.9$) |
| $\lambda_{\mathrm{lb}}$ | Linear | 4 | 2 | 0.10 | T | 80.0($+1.8$) |
| Gumbel noise | Linear | 4 | 2 | 0.01 | F | 78.1($-0.6$) |
| Top-k | Linear | 4 | 1 | 0.01 | T | 78.3($-0.4$) |
| Top-k | Linear | 4 | 3 | 0.01 | T | 79.5($+1.2$) |
| Top-k | Linear | 4 | 4 | 0.01 | T | 81.1($+3.7$) |

We select a single, fixed MoA configuration via ablations on AML-Hehr (Table 2) and apply to the rest of the dataset. Compared to the single-aggregator baseline (78.6 balanced accuracy), mixtures only help when routing is lightly regularized and stochastic. Performance is best with an intermediate number of aggregators and mild load balancing: a router with four aggregators, $\lambda_{\mathrm{lb}} = 0.01$, and Gumbel noise enabled achieves 81.5 balanced accuracy on AML-Hehr (+3.7% vs. base). Stronger regularization ($\lambda_{\mathrm{lb}} = 0.10$), disabling Gumbel noise, or changing the number of aggregators (2 or 6) reduces performance. As expected, top-1 routing achieves similar performance to the baseline (78.3 vs. 78.6), while top-2 routing yields the largest improvement over the baseline (81.5). Increasing $k$ beyond 2 does not lead to further gains. We therefore adopt the 4-aggregator configuration with top-2 routing, Gumbel noise, and $\lambda_{\mathrm{lb}} = 0.01$ as the fixed training recipe for all subsequent experiments, where it generalizes well across organs and modalities (full ablations in Appendix Table 4).

### 4.5. Limitations

We acknowledge several limitations of our study. First, as shown in prior work, no single MIL architecture uniformly outperforms all others; performance is strongly dataset dependent. In our experiments, mixtures of aggregators improve over their single-aggregator counterparts, but they may not always achieve the best performance compared to alternative MIL architectures. Second, we experimented ABMIL and TransMIL in our pipeline, future work can incorporate additional MIL backbones within the MoA framework and benchmark them against their corresponding single-aggregator baselines. Finally, although we used a fixed training recipe across all datasets, additional hyperparameter tuning may be necessary to benefit of MoA in other settings.

## 5. Conclusion

We propose Mixture of Aggregators (MoA), a framework that employs multiple aggregators for multiple instance learning to better capture diverse distributions in heterogeneous datasets. Our approach improves diagnostic performance in both pathology and hematologic cytology. Within this framework, aggregators specialize in different disease types, and each specialized aggregator provides distinct, clinically relevant attention distributions over the instances. These complementary attention patterns enhance diagnostic accuracy when combined. The code is available at https://github.com/fatihOzlugedik/MixtureOfAggregators

### Acknowledgments

CM received funding from the European Research Council under the European Union's Horizon 2020 Research and Innovation Programme (grant agreement 866411 & 101113551). We acknowledge support from the High-Tech Agenda Bayern.

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

## Appendix A. Supplementary methods

### A.1. Aggregator details

For ABMIL, we use the gated attention pooling mechanism introduced by Ilse et al (Ilse et al., 2018). Given a bag of $n$ instance embeddings $h_{i_{i=1}}^n$, the attention weight for each instance is computed as

$$a_i = \frac{\exp\big\{\, w^\top\big[\tanh(Vh_i) \odot \sigma(Uh_i)\big]\,\big\}}{\sum_{j=1}^n \exp\big\{\, w^\top\big[\tanh(Vh_j) \odot \sigma(Uh_j)\big]\,\big\}}, \tag{3}$$

where $V, U \in \mathbb{R}^{l \times d}$ and $w \in \mathbb{R}^l$ are learnable parameters, $\tanh(\cdot)$ and $\sigma(\cdot)$ denote the hyperbolic tangent and sigmoid activations, and $\odot$ is the element-wise product. The bag-level representation is then obtained as a weighted sum of instances.

$$z = \sum_{i=1}^n a_i h_i. \tag{4}$$

For Transformer archicture, we use recipe by (Wagner et al., 2023). Self attention is defined as:

$$SA(Q, K, V) = \text{softmax}\left(\frac{QK^\top}{\sqrt{d_k}}\right)V, \tag{5}$$

where queries $Q \in \mathbb{R}^{n \times d_k}$, keys $K \in \mathbb{R}^{n \times d_k}$, and values $V \in \mathbb{R}^{n \times d_v}$ are obtained from input embeddings $x$ via

$$Q = W_Q x, \quad K = W_K x, \quad V = W_V x, \tag{6}$$

with learnable weights $W_Q \in \mathbb{R}^{d \times d_k}$, $W_K \in \mathbb{R}^{d \times d_k}$, and $W_V \in \mathbb{R}^{d \times d_v}$.

### A.2. Training details

For model training, we use a fixed learning rate of $5 \times 10^{-5}$ with the AdamW optimizer (weight decay = 0.01), update gradients every 16 patients, and train for 150 epochs with early stopping based on the validation loss. We employ four aggregators and adopt a staged training schedule: during the first three epochs, all four aggregators are used equally, after which we gradually reduce the active expert count to two. The goal of this warm-up phase is to ensure that all aggregators acquire a minimal, shared understanding of the task before the router begins to decide which aggregators are most relevant for each sample. Enforcing top-2 routing encourages the aggregators to specialize rather than collapsing into a simple ensemble.

We used single H100 80GB GPU for running the experiments.

### A.3. Dataset details

**AML-Hehr** (Hehr et al., 2023) includes 189 patients from four acute myeloid leukemia (AML) genetic subtypes (PML–RARA fusion, NPM1-mutation, CBFB–MYH11 fusion and RUNX1–RUNX1T1 fusion) and healthy controls. Each patient has an average of $430 \pm 107$ single-cell white blood cell images. We hold out 43 patients as the test set.

**cAItomorph** data (Dasdelen et al., 2026) includes 2,043 patients spanning seven hematologic conditions–acute myeloid leukemia (AML), myelodysplastic syndromes (MDS), myeloproliferative neoplasms (MPN), MDS/MPN overlap syndromes, lymphoma, plasma cell neoplasms, and reactive changes–along with a healthy cohort. The number of white blood cell images per patient ranges from 55 to 500, with an average of $488 \pm 55$ cells. A total of 409 patients are held out for testing. This dataset is particularly challenging due to substantial inter-class heterogeneity and intra-class overlap.

**BCNB** (Xu et al., 2021) includes 1058 core needle biopsy slides from early breast cancer patients. It includes binary prediction tasks of ER, PR and HER2 status.

**BRACS** (Brancati et al., 2022) consist of 547 breast tissue biopsy slides from 189 patients. It includes 7 classes: normal, pathological benign, usual ductal hyperplasia, flat epithelial atypia, qtypical ductal hyperplasia, ductal carcinoma in situ and invasive carcinoma. We randomly sampled patches from the dataset to improve computational efficiency.

**CPTAC-CCRCC** (National Cancer Institute Clinical Proteomic Tumor Analysis Consortium (CPTAC), 2018a) has 103 slides with clear cell renal cell carcinoma. We predict the immune class of patients: low, medium, high.

**CPTAC-HNSCC** (National Cancer Institute Clinical Proteomic Tumor Analysis Consortium (CPTAC), 2018b) has 107 slides with head and neck cancer. Immune class prediction is made (low, medium, high).

**CPTAC-LSCC** (National Cancer Institute Clinical Proteomic Tumor Analysis Consortium (CPTAC), 2018c) includes 104 slides with lung squamous cell carcinoma. Histologic grade of the patients are predicted (well differentiated vs moderately differentiated)

**CPTAC-UCEC** (National Cancer Institute Clinical Proteomic Tumor Analysis Consortium (CPTAC), 2019) has 94 slides with endometrial carcinoma. Immune class prediction is made (low, medium, high).

**HANCOCK** (Dörrich et al., 2025) is a multimodal dataset of 763 head and neck cancer patients. We include 6 different task from this dataset. 5 of them are binary including: Keratinizing squamous cell carcinoma grading (n=383), non keratinizing squamous cell carcinoma grading (n=74), lymphovascular invasion (n=697), perineural invasion (n=697) and primary vs. metastasis tumor (n=676). Prediction of primary tumor site includes 696 cases from 4 different location (oral cavity, larynx, oropharynx, hypopharynx)

**IMP-Cervix** (Oliveira et al., 2024) includes 5333 samples from cervical biopsy. Cervical cancer grade is predicted: non-neoplastic, low-grade, high-grade.

For train-test splits, we follow PathoBench benchmark (Zhang et al., 2025). We held out the test split and apply 5 cross-validation within training set.

## Appendix B. Supplementary findings

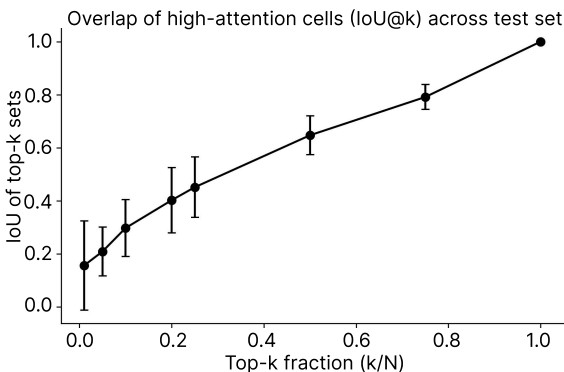

Figure 4: In AML-Hehr, we quantify the overlap between the most-attended cells selected by different aggregators using the intersection over union (IoU) of their top-$k$ sets, computed for each test sample and then averaged across the test set (mean $\pm$ s.d.). We evaluate $k$ as a fraction of cells per patient ($k/N \in \{0.01, 0.05, 0.10, 0.20, 0.25, 0.50, 0.75, 1\}$). IoU increases with $k$, but remains low for small $k$ (e.g., IoU@0.01 $0.16 \pm 0.17$, IoU@0.05 $0.21 \pm 0.10$), indicating that aggregators prioritize largely distinct subsets of high-attention cells.

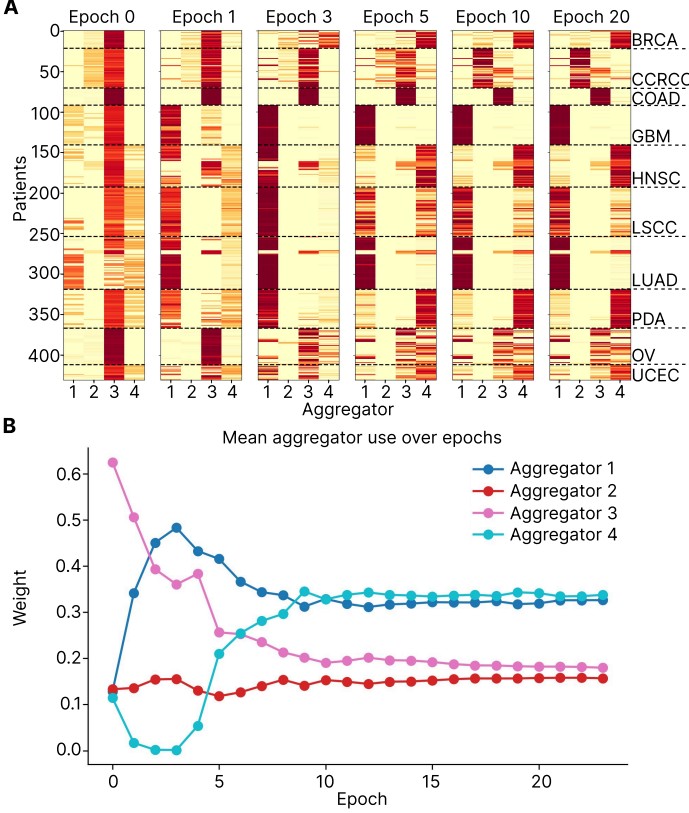

Figure 5: Aggregator specialization over training epochs on CPTAC organ classification. We train MoA with four aggregators on the CPTAC organ classification task and visualize how the router's per-sample weights evolve during optimization. (A) Heatmaps show router-assigned weights for each patient (rows; grouped by organ type, dashed separators) across aggregators (columns) at selected epochs (0, 1, 3, 5, 10, 20). Early in training (epochs 0–1), routing is unstable and can be dominated by a single aggregator due to random initialization. As training progresses (epochs 3–20), organ-specific routing patterns emerge, indicating specialization of different aggregators for different organ types and increased utilization of multiple aggregators. (B) Mean router weight per aggregator across all samples over epochs. After the initial transient phase, the weights stabilize, with aggregators 1 and 4 receiving higher average weight than the others, reflecting more frequent use across samples, while aggregators 2 and 3 are used less on average.

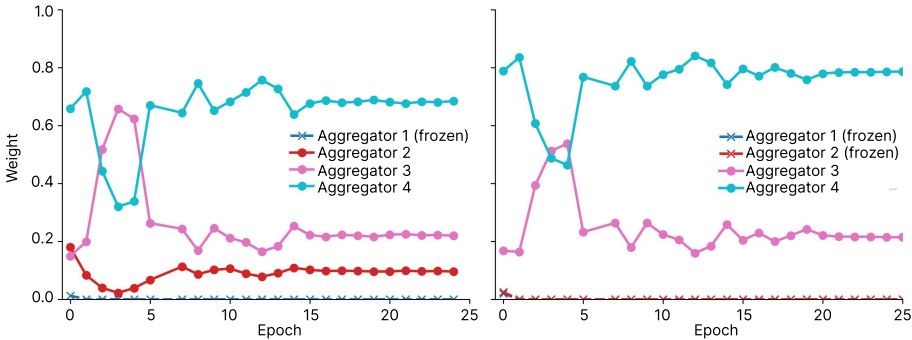

Figure 6: Aggregator corruption analysis on HANCOCK. To test whether the router can suppress underperforming aggregators, we intentionally created "corrupted" aggregators by freezing the weights of $k = 1$ (left) and $k = 2$ (right) aggregators during training, while keeping the remaining aggregators trainable. We visualize the router's weight distribution over epochs. In both settings, the router rapidly discards the frozen aggregators and reallocates probability mass to the trainable ones, indicating that it can identify and effectively eliminate poor aggregators early in training

Table 3: Performance (mean ± std, in %) for mixed-aggregator MoA (2×ABMIL + 2×TransMIL) compared with pure MoA ABMIL / MoA TransMIL and their baselines.

| Dataset / Task | ABMIL | TransMIL | ABMIL MoA | TransMIL MoA | Mixed MoA |
|---|---|---|---|---|---|
| AML-Hehr | 81.5 ± 3.7 | 78.6 ± 2.2 | 78.4 ± 2.2 | 81.5 ± 1.0 | 80.1 ± 3.2 |
| cAItomorph | 50.9 ± 1.9 | 59.4 ± 1.9 | 52.8 ± 1.8 | 60.1 ± 1.1 | 58.6 ± 1.6 |
| BCNB/ER | 91.2 ± 0.4 | 85.7 ± 3.3 | 91.3 ± 0.2 | 88.4 ± 0.5 | 90.6 ± 0.9 |
| BCNB/HER2 | 84.0 ± 0.5 | 69.8 ± 3.5 | 84.0 ± 0.7 | 81.2 ± 2.2 | 78.3 ± 2.3 |
| BCNB/PR | 88.2 ± 0.6 | 78.2 ± 3.5 | 88.3 ± 0.4 | 84.7 ± 0.7 | 87.0 ± 0.5 |
| BRACS | 34.4 ± 1.5 | 26.5 ± 1.5 | 34.6 ± 1.9 | 29.3 ± 1.6 | 35.4 ± 2.4 |
| CPTAC-ALL | 95.5 ± 0.2 | 95.5 ± 1.0 | 96.1 ± 0.3 | 96.3 ± 0.6 | 95.9 ± 0.5 |
| CPTAC-CCRCC | 43.5 ± 4.3 | 45.2 ± 3.7 | 45.4 ± 7.7 | 47.2 ± 4.1 | 48.0 ± 2.7 |
| CPTAC-HNSC | 33.0 ± 4.6 | 27.9 ± 5.8 | 35.1 ± 5.7 | 31.6 ± 3.6 | 35.1 ± 3.1 |
| CPTAC-LSCC | 67.1 ± 3.5 | 60.0 ± 9.4 | 69.8 ± 2.2 | 63.7 ± 5.2 | 63.5 ± 4.0 |
| CPTAC-PDA / | 35.1 ± 3.0 | 32.9 ± 7.0 | 39.3 ± 7.3 | 41.3 ± 4.1 | 41.3 ± 2.3 |
| CPTAC-UCEC | 36.3 ± 5.6 | 29.7 ± 7.9 | 43.2 ± 2.8 | 44.9 ± 7.3 | 42.1 ± 3.1 |
| HANCOCK/K-SCC grading | 71.4 ± 5.8 | 60.8 ± 6.6 | 73.8 ± 1.3 | 73.6 ± 2.9 | 73.0 ± 1.7 |
| HANCOCK/NK-SCC grading | 62.0 ± 10.8 | 48.0 ± 9.7 | 67.0 ± 5.8 | 61.0 ± 10.2 | 59.0 ± 10.2 |
| HANCOCK/perineural invasion | 76.9 ± 0.7 | 63.9 ± 5.6 | 79.8 ± 1.1 | 75.5 ± 3.2 | 76.7 ± 3.0 |
| HANCOCK/metastasis | 71.4 ± 1.7 | 63.2 ± 5.5 | 74.8 ± 1.3 | 64.7 ± 3.6 | 68.2 ± 2.4 |
| HANCOCK/tumor site | 68.5 ± 1.9 | 66.7 ± 1.0 | 74.1 ± 3.2 | 71.8 ± 3.2 | 72.2 ± 2.4 |
| HANCOCK/vascular invasion | 51.6 ± 7.6 | 59.9 ± 6.0 | 55.3 ± 6.8 | 66.8 ± 3.8 | 66.1 ± 5.1 |
| IMP-Cervix | 45.0 ± 3.6 | 52.9 ± 4.5 | 46.6 ± 2.8 | 57.0 ± 1.9 | 53.4 ± 3.0 |

Table 4: Full ablation of different configurations on AML-Hehr data. *Base* = baseline (single aggregator, no router); *Lin.* = Linear; *MLP* = multi-layer perceptron. Second line shows number of experts used. Third line for load balancinf loss coeff $\lambda_{lb}$; fourth line Gumbel routing (True/False). **Bold** indicates the best-performing configuration

| router-arch: | Base | MLP | Lin | Lin | MLP | MLP | Lin | Lin | MLP | MLP | Lin | Lin | MLP | Lin | MLP | MLP | Lin | Lin | MLP | MLP | **Lin** | Lin | MLP | MLP | Lin |
|---|---|---|---|---|---|---|---|---|---|---|---|---|---|---|---|---|---|---|---|---|---|---|---|---|---|
| #Experts: | – | 2 | 2 | 2 | 2 | 4 | 4 | 4 | 4 | 6 | 6 | 6 | 6 | 2 | 2 | 2 | 2 | 4 | 4 | 4 | **4** | 6 | 6 | 6 | 6 |
| $\lambda_{lb}$: | – | 0.01 | 0.01 | 0.10 | 0.10 | 0.01 | 0.01 | 0.10 | 0.10 | 0.01 | 0.01 | 0.10 | 0.10 | 0.10 | 0.01 | 0.10 | 0.01 | 0.10 | 0.01 | 0.10 | **0.01** | 0.10 | 0.01 | 0.10 | 0.01 |
| Gumbel: | – | F | F | F | F | F | F | F | F | F | F | F | F | T | T | T | T | T | T | T | **T** | T | T | T | T |
| Balanced Acc | 78.6 | 79.1 | 78.5 | 78.5 | 79.1 | 74.9 | 78.1 | 75.8 | 78.5 | 79.1 | 78.1 | 77.6 | 72.2 | 76.8 | 79.6 | 77.8 | 76.8 | 80.0 | 76.1 | 76.3 | **81.5** | 74.1 | 78.0 | 78.2 | 79.3 |
| % vs Base | – | +0.6 | -0.1 | -0.1 | +0.6 | -4.7 | -0.6 | -3.6 | -0.1 | +0.6 | -0.6 | -1.3 | -8.1 | -2.3 | +1.3 | -1.0 | -2.3 | +1.8 | -3.2 | -2.9 | **+3.7** | -5.7 | -0.8 | -0.5 | +0.9 |

Table 5: Inference times of different MIL architectures on the HANCOCK dataset. MoA increases inference time by approximately $2.5\times$ compared to a single-aggregator model.

| Architecture | Total time [s] | Time / sample [ms] | # of parameters |
|---|---|---|---|
| ABMIL | 0.052 | 0.368 | 0.69 M |
| MoA-ABMIL | 0.148 | 1.052 | 3.15 M |
| TransMIL | 0.462 | 3.274 | 2.54 M |
| MoA-TransMIL | 1.100 | 7.805 | 10.56 M |
| DSMIL | 0.101 | 0.716 | 0.86 M |
| CLAM-SB | 0.198 | 1.401 | 0.79 M |
| Mean | 0.025 | 0.173 | 0.43 M |

Table 6: Comparison of MoA with ensemble baselines on AML-Hehr. We compare a single-aggregator baseline with MoA using top-2 routing and with two ensemble baselines that use all four aggregators without routing. We evaluate both averaging the aggregator outputs (mean ensemble) and concatenation (concat ensemble). Median inference time is reported per sample, along with the relative slowdown compared with the single-aggregator baseline. MoA achieves the best balanced accuracy $(81.5 \pm 1.0)$ while requiring only a $2.05\times$ slowdown. Mean ensemble achieves similar performance with higher inference time.

| Method | # aggregators | Balanced Acc. | Median inference time (ms / sample) | Relative slowdown vs. single |
|---|---|---|---|---|
| Baseline | 1 | $78.6 \pm 2.2$ | 1.783 | $1.0\times$ |
| MoA (top-2 routing) | 4 | $81.5 \pm 1.0$ | 3.663 | $2.05\times$ |
| Mean Ensemble | 4 | $81.3 \pm 1.2$ | 6.343 | $3.56\times$ |
| Concat Ensemble | 4 | $78.3 \pm 2.2$ | 6.330 | $3.55\times$ |

