# OpenReview forum: "MoA: Mixture of Aggregators Improves Slide-Level Diagnosis in Computational Pathology"
_MIDL.io/2026/Conference — MIDL 2026 Poster_

### Official Review · Reviewer_Xm8Q · 2026-01-08

**Confidence:** 4
**Preliminary Rating:** 2
**Final Rating:** 3

**Summary:**

The authors provide a framework of a Mixture of Aggregators (MoA) of Whole Slide Image analysis, an approach for Multiple Instance learning (MIL).  Over classical attention-based MIL, which utilizes the same feature extraction and projection for all patches, they describe a diversified projection strategy. With the MoA approach it is possible that the framework captures diverse morphological patterns in heterogeneous pathology and cytology data. They perform extensive tests on both domains.

**Strengths:**

- The authors present a nice lean adaptation of the Mixture‑of‑Experts framework for Whole‑Slide‑Image (WSI) analysis.
- Their extensive empirical validation - spanning 19 datasets and employing a 5‑fold cross‑validation protocol underscores the robustness of the proposed approach.

**Weaknesses:**

- I am not really convinced by the novelty claim of the authors. Applying MoE on WSIs was already shown to be effective by Li et al. (Medical Image Analysis, Volume 103, July 2025, 103561, available online since April 2025, https://doi.org/10.1016/j.media.2025.103561). In this work, entitled “M4: Multi-proxy multi-gate mixture of experts network for multiple instance learning in histopathology image analysis”, the authors propose a MoE approach that is applied in much the same way than the authors do, but for a multi label use case with different aggregators/routers for each class. Given that this paper is essentially about showing a new methods, it would be a shame if this method wasn't new.

- The authors state that "In essence, MoA is not a mere import of MoE into pathology; rather, it re-designs the aggregation stage to reflect the complex nature of histological evidence, bridging the gap between single-aggregator MIL and the heterogeneous reasoning pathologists employ in practice." - but I'm really not sure what the difference is here. How is it not a (seemingly straight-forward, but also sensible) application of the MoE principle (with routers and the aggregators being in the role of the experts) to MIL?

- The authors write that they use k=2 for the top-k routing. I didn't find how this value was established, and also saw no ablation. At the very least, I would recommend a motivation for it.

- It is unclear if the feature extractors (UNI, DinoBloom-B) are the same in the experiments of ABMIL, TransMIL, CLAM-SB, DSMIL and MeanMIL. To provide comparable results, they should necessarily be, but it's not clear from the manuscript.

**Detailed Comments:**

- Fig 1.: Please describe the functionality of the aggregators.
- Please specify whether any augmentation strategies (e.g., rotation, flipping, color jitter) or other pre‑processing steps (normalisation, stain‑normalisation, etc.) were applied, and describe the implementation details.
- Provide the full list of hyper‑parameters  / optimizer / scheduler used during training (Batch-size, learning rate, image size, etc.)
- A link to the codebase (or a statement that the code will be released upon acceptance) would greatly aid reproducibility and allow the community to build upon your work.

**Justification Of Final Rating:**

I stand by my criticisms that the approach of this paper is basically the same as previous work by Li et al., and the only modest change is that it is being used in a 1-class rather than an N-class scenario, and that it uses multiple router heads for this 1 class scenario.

However, I think that the authors overall reacted well to the comments of the reviewers, which I why I am upgrading my rating. I still think that it is misleading that the authors try to frame the MoE approach as "mixture of aggregators", while this is clearly also what Li et al. did in their work, hence, I am only upgrading to "borderline".

**Justification Of The Preliminary Rating:**

I think the major problem of this work is that the method is not new but was already shown to work well for a very similar use case. Since science is also about reproduction, I would find it acceptable in principle, if the work was centered around reproduction or benchmarking, but since the authors claim to introduce a new method I can't recommend to accept the work in its current form.

**Questions To Address In The Rebuttal:**

- How is the approach not a straight-forward adaptation of MoE to MIL, as the authors claim?
- How is the approach novel, given that Li et al. have already published a very similar (or even extended) approach in the beginning of 2025 (preprint was available in July 2024).
- Please clarify if for all experiments the same feature extractors have been used.

---

> ### Author Response · Authors · 2026-01-25
>
> Dear Reviewer, we thank you for your valuable comments and suggestions to improve the paper. Here is a point-by-point response:
>
> **Novelty:** Thank you for pointing out the concurrent work of Li et al. (M4) [1]. We acknowledge their work in Section 2.2, where we now write: “Prior work has explored MoE in computational pathology primarily in multi-task learning settings, where experts are used to share representations across related tasks (e.g.,mutation prediction) via task-specific multi-gated routing (Li et al., 2025). In this study, we introduce a slide-level MoE approach that combines multiple aggregators under a single router, which we term Mixture of Aggregators (MoA). In contrast to (Li et al., 2025), our work focuses on single-task diagnosis and aims to capture slide-level heterogeneity.”
>
> Acknowledging this point, we also removed the sentence starting with: *“In essence, MoA is not a mere import of MoE into pathology…"*
>
> We also agree that our approach is inspired by MoE architectures, where experts are typically feed-forward networks or subnetworks that perform task or token-level feature transformations, with routing primarily motivated by computational efficiency or capacity scaling [2]. Although motivation is the same, we would like to point out that we transfer the  MoE principle to the aggregation stage and explore benefits and downsides. Our router predicts sample-specific attention distributions over instances, selecting among multiple attention-based aggregation operators. Each expert therefore represents a distinct pooling strategy,  allowing the model to adapt how evidence is aggregated for each sample.
>
> **Top-k routing selection:** Thank you for pointing out the choice of top-2 routing. Following your suggestion, we conducted ablations for k = 1, 2, 3, and 4 (max) and added in Table 2. The following text is inserted in Section 4.4: “As expected, top-1 routing achieves similar performance to the baseline (78.3 vs. 78.6), while top-2 routing yields the largest improvement over the baseline (81.5). Increasing k beyond 2 does not lead to further gains.”
>
> Although top-2 aggregators yielded the best results with four aggregators in our ablations, the number of aggregators and the choice of top-k can be adjusted depending on the dataset and the task.
>
> **Backbone extractors in the experiments:** As mentioned in Section 4.1, we used UNI [3] to embed all pathology datasets and DinoBloom-B [4] for the hematologic cytology datasets in all aggregator experiments.
>
> **Training details for each aggregator:** We did not use any augmentation strategy, as many augmentation methods apply at the instance level, whereas we evaluate our approach at the patient level. For each feature extractor, we followed the recommended image resizing and normalization (224×224 image size and ImageNet normalization values for both UNI and DinoBloom models).
> As mentioned in Section A.2 (Training details), we used the same training recipe for all aggregators.
>
> **Code for linkbase:** Along with the paper, we provided the codebase for training and inference at https://github.com/fatihOzlugedik/MixtureOfAggregators.
>
> **Ref:**
>
> [1] Li, Junyu, et al. "M4: Multi-proxy multi-gate mixture of experts network for multiple instance learning in histopathology image analysis." Medical Image Analysis 103 (2025): 103561.
>
> [2] Shazeer et al. “Outrageously Large Neural Networks: The Sparsely-Gated Mixture-of-Experts Layer”
>
> [3] Chen, Richard J., et al. "Towards a general-purpose foundation model for computational pathology." Nature medicine 30.3 (2024): 850-862.
>
> [4] Koch, Valentin, et al. "DinoBloom: a foundation model for generalizable cell embeddings in hematology." MICCAI. Cham: Springer Nature Switzerland, 2024.

---

> > ### Comment · Reviewer_Xm8Q · 2026-01-28
> > **Comment to the authors**
> >
> > Thank you for your rebuttal document and also for updating your paper, incorporating the 2024/2025 work by Li et al.
> >
> > - I think to call their work concurrent is honestly more than a bit of a stretch. The preprint came out in July 2024, and the authors can hardly argument that the authors did not choose proper keywords. It was literally the first hit on google scholar for me, searching for "mixture of experts multiple instance learning".
> > - Besides, I still feel that the work auf the authors is a specialization of the general concept of the Li et al. MedIA paper, but maybe the authors can help my understanding here. Li et al. use a feature extractor on WSIs (like in the original Ilse et al. work), and then have multiple sub-networks ("experts") that extract task-related information before aggregation, and then have multiple routers that gate (or weight) the expert's outputs, before a final task-related network mapping to the final output. If I take the special case of n=1 task networks, I'm getting pretty much exactly the same approach described by the authors of this MIDL paper. Maybe the authors could comment on what I'm missing here. Don't get me wrong here, I think large-scale reevaluations of previously published paradigms do have merits in science. But If my assumptions are right, I think that claiming novelty on a specialization / reduction of a previously published principle is a bit off.
> > - Thank you for running ablations on the choice of k for top-k-routing.
> > - Thank you for providing a repository with the code. Upon inspection, I found it a bit hard to navigate. I recommend to add more explanations here. A "getting started" would tremendously help. For instance, there is no mention of how to generate the required h5 files with patch embeddings (using the CLAM pipeline?).

---

> > > ### Author Response · Authors · 2026-01-30
> > >
> > > We thank the reviewer for the detailed assessment of our revision and would like to comment on the issues raised.
> > >
> > > **Regarding previous work:** We agree that the overall architecture is similar to the work of Li et al., [1] and that scientific progress is often incremental and confirmatory. Notably, our work is inspired by MoE-style ideas, but we demonstrate that our aggregation-level routing strategy is effective across different domains, tasks, and datasets, and we provide additional analyses of specialization and routing behavior in these settings. Let us outline the two key differences that we believe make our approach stand out:
> > >
> > > (i) The “experts” in Li et al. are not pure aggregators. Each expert includes an additional feature transformation/extraction stage on top of the patch embeddings and then performs aggregation. In our approach, all aggregators operate on the same shared embedding space and differ only in how they leverage the instance distribution. In other words, our goal is to learn multiple complementary readouts of the same bag rather than generate multiple bags. Practically, our method stays within the standard computational pathology MIL pipeline (see MIL-LAB [2]) and introduces two additions:  replacing a single aggregator with multiple aggregators, and adding a router to adaptively combine them. This also enables a plug-and-play design, allowing you to easily swap the feature extractor and the aggregator architecture.
> > >
> > > (ii) Li et al.’s method is primarily designed for multi-label/multi-task learning, which may not be available in every dataset setting or clinical problem, whereas our method is explicitly designed for single-task prediction. Architecturally, as you highlighted, Li et al. use $N$ experts and $N$ routers/gates, whereas our pipeline uses $N$ aggregators with a shared router.
> > >
> > > **Codebase:** Thank you for reviewing the codebase. Following your advice, we polished the README to improve navigation. As mentioned in the paper, the HDF5 (h5) files were generated using the TRIDENT [3] pipeline, and we now also add this information to the repository.
> > >
> > > Ref:
> > >
> > > [1] Li, Junyu, et al. "M4: Multi-proxy multi-gate mixture of experts network for multiple instance learning in histopathology image analysis." Medical Image Analysis 103 (2025): 103561.
> > >
> > > [2] https://github.com/mahmoodlab/MIL-Lab
> > >
> > > [3] https://github.com/mahmoodlab/TRIDENT

---

> > > > ### Comment · Reviewer_Xm8Q · 2026-02-02
> > > > **Figure 1 now missing**
> > > >
> > > > Dear authors,
> > > >
> > > > while revisiting your uploaded pdf, I realized that the main architectural figure (Fig. 1) is now missing (or, to be more precise: The graphic is missing, the caption is still there). I am sure this was a mishap, but wanted to let you know so you can rectify this.
> > > >
> > > > (1) Regarding the difference between MoE and MoA: I still don't get it. I reviewed the Li et al. paper once more, and they clearly use the same feature extractor for all patches (see Figure 2), followed by MLPs acting as "experts", but essentially calculating a projection to a latent space where weight-and-sum-based aggregation based on the weights of the router can be done. As far as I understand the work of the authors, the "aggregators" are also permutation-invariant projections realized by an MLP and later weighted by the router.
> > > >
> > > > Thank you for updating the README.
> > > >
> > > > (2) Stating that the Li et al. method is designed for multi-class scenarios only is a bit of a deterring argument. While I agree with the authors that they essentially have a single head per class with N classes, the addition to multiple heads is not a major architectural addition.

---

> > > > > ### Author Response · Authors · 2026-02-02
> > > > >
> > > > > Dear Reviewer,
> > > > >
> > > > > Thank you for pointing out the rendering issue. We have checked the manuscript and were unable to reproduce the problem with Figure 1 (tested in macOS Preview, Adobe Reader, and Google Chrome). Could you please let us know which operating system and PDF viewer/platform you are using?

---

> > > > > > ### Comment · Reviewer_Xm8Q · 2026-02-02
> > > > > > **Re: Rendering issue**
> > > > > >
> > > > > > Dear authors,
> > > > > >
> > > > > > thanks for your fast response. That's weird indeed. I can reproduce the issue on Safari 26.2 (i.e., without downloading the paper, just by clicking on the link), but not on Preview/Mac OS 26.2.

---

> > > > > > > ### Author Response · Authors · 2026-02-02
> > > > > > > **Re: Rendering issue**
> > > > > > >
> > > > > > > Dear Reviewer,
> > > > > > >
> > > > > > > Thank you for letting us know. We have now also observed the same issue in Safari (v17.5). The figures are embedded in PDF format, and it appears that Safari is unable to render them properly. Converting the figures to PNG resolves the problem. We will ensure that the figures are displayed correctly across platforms

---

### Official Review · Reviewer_u2Xe · 2026-01-08

**Confidence:** 4
**Preliminary Rating:** 3
**Final Rating:** 4

**Summary:**

The authors propose "Mixture of Aggregators" (MoA), a framework incorporating the Mixture-of-Experts (MoE) paradigm into the aggregation layer of Multiple Instance Learning (MIL) for computational pathology. Instead of relying on a single pooling mechanism (e.g., ABMIL or TransMIL), the method employs a pool of aggregators (experts) and a router network. The router dynamically assigns slide-level bags to the top-2 most relevant aggregators using a gating mechanism with load-balancing loss and Gumbel noise. The outputs are then fused for the final prediction. The method is evaluated on 19 tasks across 16 datasets, including both histology and cytology, using UNI and DinoBloom feature extractors. Results show an average performance improvement of 4.5% over ABMIL and 12.6% over TransMIL.

**Strengths:**

- Extensive Evaluation: The authors conduct a comprehensive evaluation across a diverse set of 19 tasks, spanning both histology (e.g., BRACS, CPTAC) and cytology (AML-Hehr). This breadth demonstrates the generalizability of the proposed framework across different tissue types and magnifications.
- Plug-and-Play Design: The MoA framework is agnostic to the underlying aggregator architecture. The paper demonstrates this by applying it to both ABMIL and Transformer-based aggregators, making it a flexible extension to existing MIL pipelines.
- Interpretability Analysis: The paper attempts to open the "black box" of the router by visualizing which aggregators attend to which morphological patterns (Figure 3). The use of Jensen-Shannon divergence to quantify the difference in attention distributions between aggregators is a thoughtful addition.
- Ablation Studies: The authors provide a decent ablation study regarding the routing mechanism (load balancing $\lambda_{lb}$ and Gumbel noise), justifying their specific design choices for the gating mechanism.

**Weaknesses:**

- Incremental Novelty: The core contribution is the straightforward application of the well-known Mixture-of-Experts (MoE) paradigm (Shazeer et al., 2017) to the MIL aggregation head. While effective, this is a relatively standard architectural modification rather than a fundamental shift in how slide-level representations are learned. The conceptual leap from "Single Aggregator" to "MoE Aggregator" is logical but lacks deep theoretical innovation specific to pathology.
- Lack of "Naive Ensemble" Baseline: The paper claims the router is key to selecting specialized experts. However, it is not clear if the performance gain comes from the dynamic routing or simply from having more parameters (ensemble effect). A critical missing baseline is a "Naive Ensemble" where the outputs of all $K$ aggregators are simply averaged (or concatenated) without a router. If a static ensemble achieves similar performance, the complexity of the routing mechanism and load-balancing losses is unnecessary.
- Homogeneity of Experts: In the experiments, the "mixture" consists of multiple instances of the same architecture (e.g., 4 $\times$ ABMIL). True MoE gains often arise from experts that are fundamentally different or specialized (e.g., combining a local-texture expert with a global-context expert). Using identical architectures initialized differently relies on stochasticity to induce specialization, which feels less robust.Computational Cost vs. Clinical Benefit: Table 4 reveals a significant computational overhead (approx. $2.5\times$ increase in inference time). In the context of recent "Green AI" and clinical deployment trends where latency matters, a $2.5\times$ cost for a marginal gain (e.g., +0.1% on BCNB/ER, +0.6% on BRACS with ABMIL) is hard to justify.
- Questionable Specialization Claims: While Figure 3 shows different attention maps, it is unclear if this "specialization" is semantically consistent across the dataset or just random variations. The paper claims aggregators focus on "distinct disease-specific distributions," but without a mechanism to enforce semantic disentanglement (like orthogonal constraints), this interpretation might be over-fitting to the visualized samples.

**Detailed Comments:**

- Clarification on Parameter Count: Please explicitly state the total parameter count of the MoA models compared to the baselines. Is the comparison fair in terms of model capacity?
- Inference Latency: The $2.5\times$ slowdown is significant. The authors should discuss if this latency is due to the sequential processing or if it can be parallelized. Furthermore, does the router computation itself add significant overhead, or is it purely the execution of multiple aggregators?
- Stability: MoE training is notoriously unstable. Did the authors experience mode collapse where the router trivially selects only one aggregator despite the load-balancing loss? A plot of expert utilization over training epochs would be beneficial.
- Baseline Comparisons: The baselines (CLAM, DSMIL, TransMIL) are standard but slightly dated. More recent state-of-the-art methods involving long-context modeling (e.g., Mamba-based MIL) or graph-based approaches are missing. Comparing against a stronger, modern single-model baseline would strengthen the paper.
- Figure 3 Interpretation: In Figure 3A, Aggregator 2 and 4 seem to attend to different cells. Is there a quantitative metric to measure the overlap of high-attention regions across the entire test set (e.g., Intersection over Union of top-k patches) to prove this is a systematic behavior and not anecdotal?

**Justification Of Final Rating:**

While the methodological shift from single aggregators to an MoE framework is relatively incremental, the authors successfully demonstrated through the rebuttal that MoA achieves ensemble-level performance with significantly better computational efficiency, and their quantitative IoU analysis provides objective evidence of expert specialization; given the exhaustive evaluation across 19 tasks and the proven robustness of the routing mechanism, the paper represents a solid, practically valuable contribution to computational pathology and warrants acceptance.

**Justification Of The Preliminary Rating:**

The paper presents a solid execution of a logical idea (MoE for MIL) with extensive experimentation. However, the novelty is limited, and the method behaves more like a complex ensemble than a distinctively new approach to pathology. The lack of a simple ensemble baseline makes it difficult to attribute the gains to the proposed routing mechanism. The computational overhead is also a concern for practical adoption. The paper is "borderline" because while the results are positive, the technical contribution is incremental and the analysis of why it works (vs. simple ensembling) is not fully convincing.

**Questions To Address In The Rebuttal:**

- Ensemble vs. Mixture: Can you provide results for a "Naive Ensemble" baseline (averaging the outputs of the 4 aggregators without routing)? This is crucial to establish that the dynamic routing is actually adding value over a simple ensemble.
- Expert Heterogeneity: Have you experimented with mixing different types of aggregators (e.g., 2 ABMIL + 2 TransMIL) within the same MoA? This would theoretically leverage the strengths of both inductive biases.
- Cost-Benefit Ratio: Given the $2.5\times$ inference cost, how do you justify the trade-off for tasks where the improvement is $<1\%$? Is there a "lite" version of MoA possible?
- Router Generalization: Does the router learn to ignore "bad" aggregators? If one aggregator is intentionally initialized poorly or corrupted, can the router successfully filter it out?

---

> ### Author Response · Authors · 2026-01-25
>
> Dear Reviewer, we thank you for your detailed comments and feedback. We addressed all points and by that, improved our manuscript considerably. Please find a point-by-point response below:
>
> **Novelty:** We agree that our approach is inspired by MoE architectures, where experts are typically feed-forward networks or subnetworks that perform task or token-level feature transformations, with routing primarily motivated by computational efficiency or capacity scaling [1]. Although motivation is the same, we would like to point out that we transfer the  MoE principle to the aggregation stage and explore benefits and downsides. Our router predicts sample-specific attention distributions over instances, selecting among multiple attention-based aggregation operators. Each expert therefore represents a distinct pooling strategy,  allowing the model to adapt how evidence is aggregated for each sample
>
> **Ensemble baseline:** We highly appreciate your suggestion for an ensemble baseline, and included a new comparison in Table 6, in which all four aggregators are used without a router. We conduct two experiments: (i) averaging the outputs of the aggregators, and (ii) concatenating their outputs. Mean ensemble achieves similar performance to MoA with top-2 routing, but with considerably higher compute time. Thus, a smart weight allocation and aggregator selection can match ensemble-level performance while reducing inference time. Notably, we observed a similar trend in the top-k router ablations where k=4 performs comparably to k=2 (see Table 2).
>
> **Other aggregator baselines:** Our motivation for including CLAM and DSMIL as baselines is that both are designed to better capture intra-bag heterogeneity and provide interpretability. We also include MeanMIL as the most basic baseline. Across 19 tasks, there is no single architecture that consistently wins; in some cases, even MeanMIL performs best. Our aim in this study is to propose a framework that uses multiple aggregators, enhancing performance and interpretability compared with a single-aggregator variant.
> We now acknowledge your point in the limitation section: “we experimented ABMIL and TransMIL in our pipeline, future work can incorporate additional MIL backbones within the MoA framework and benchmark them against their corresponding single-aggregator baselines”
>
> **Attention distribution analysis:** Thank you for suggesting a way to show attention differences. Following your comment, we added another quantitative evaluation for the AML-Hehr task in Figure 4 and mentioned it in the text as follows: “For quantitative evaluation, we calculated the intersection over union (IoU) between the high-attention top-k fraction of cells selected by different aggregators (Appendix Figure 4). The mean IoU across test samples is 0.16±0.17 at k=0.01 and 0.21±0.10 at k=0.05, showing that only a small fraction of highly attended cells are shared between aggregators.”
>
> **Aggregator specialization over time:** Thank you for pointing out aggregator specialization over time. We have now added Figure 5 to the Appendix, showing aggregator specialization on the CPTAC dataset, which has more classes and higher heterogeneity, and where all aggregators are used.
>
> **Parameter count and inference time:** Thank you for raising the concern regarding latency and computational burden in the context of green AI. In Appendix Table 5, we now appended another column showing the number of parameters. The main advantage of the mixture of aggregators is that it has 4 times more parameters than a single aggregator, while requiring only 2.5x more inference time.
> We agree that, for some datasets, the gains are quite marginal given the increased inference time (BCNB, +0.1%). Regardless of the improvement, we report results for all datasets used for testing to ensure transparency. On the other hand, the improvement of MoA is substantial in CPTAC-PDA (+25.5%) and UCEC (+51.2%), and moderate to high in LSCC (+4.0%), HANCOCK K-SCC grading (+3.4%),  NK-SCC grading (+8.1%), perineural invasion (+3.8%), metastasis (+4.8%), and tumor site (+8.2%).
>
> **Mixing different aggregators:** Thank you for this compelling idea. We tried mixing two ABMIL and two TransMIL models as aggregators. The results are provided in Appendix Table 3 and mentioned in the text as follows: “The mixture of ABMIL and TransMIL aggregators within the MoA framework (2xABMIL + 2xTransMIL) does not provide additional benefit and often performs between the pure ABMIL MoA and pure TransMIL MoA”.
>
> **Aggregator corruption analysis:** As you suggested, we created a poor aggregator and tested whether it is eliminated during training. After randomly initializing each aggregator, we froze aggregators k=1,2 and ran training. The router effectively learns to discard the corrupted aggregator in the early epochs (Figure 6).
>
> **Ref:**
> [1] Shazeer et al. “Outrageously Large Neural Networks: The Sparsely-Gated Mixture-of-Experts Layer”

---

> > ### Comment · Reviewer_u2Xe · 2026-01-28
> >
> > After reading the rebuttal, I acknowledge that the authors have addressed several of my key concerns in a substantive way. In particular, the addition of a “naive ensemble” baseline (averaging or concatenating the outputs of multiple aggregators without routing) is very helpful: it shows that simple ensembling can indeed reach performance comparable to the proposed MoA with top‑2 routing, but at higher computational cost, which supports the claim that routing can preserve ensemble‑level performance while reducing inference overhead. The new experiments mixing different aggregator types (2×ABMIL + 2×TransMIL) are also informative: they indicate that heterogeneity in expert architectures does not automatically yield further gains and often lands between the homogeneous MoA variants, which clarifies the limitations of expert diversity in this setting. In addition, the authors provide more detailed analyses on attention distributions (IoU of top‑k regions), expert utilization over time, parameter counts, and inference time; the corruption experiment further demonstrates that the router can learn to down‑weight a deliberately poor aggregator. These additions strengthen the empirical characterization of the method and improve transparency about the cost–benefit trade‑offs. That said, the underlying architectural idea remains a relatively straightforward adaptation of MoE to the MIL aggregation stage, and for some tasks the performance gains are modest relative to the increased complexity and latency. Overall, the rebuttal improves my understanding of where MoA is most beneficial and clarifies that its advantages are not purely due to naive ensembling, while the broader questions about methodological novelty and practical deployment trade‑offs remain to some extent.

---

> > > ### Author Response · Authors · 2026-01-30
> > >
> > > We thank the reviewer for their acknowledgment of our thorough response

---

### Official Review · Reviewer_Zn6F · 2026-01-10

**Confidence:** 3
**Preliminary Rating:** 3
**Final Rating:** 4

**Summary:**

This paper proposes a mixture-of-aggregators (MoA) framework for WSI analysis to better capture morphological heterogeneity and complementary patterns in instance distributions. MoA follows a mixture-of-experts design: it runs multiple aggregators in parallel and uses a router with top-2 gating to dynamically select and combine the most relevant aggregators for each slide. The authors evaluate the method on 19 tasks across 16 datasets, and report improved diagnostic performance, along with evidence that different aggregators learn distinct, diagnostically meaningful, and complementary instance distributions.

**Strengths:**

Applying a mixture-of-experts style approach to address heterogeneity in pathology is a natural and well-motivated strategy.

The authors conduct a comprehensive evaluation across many datasets and tasks, spanning two modalities, which suggests broad applicability. The proposed method also achieves substantial gains compared with the baselines.

**Weaknesses:**

(1) It is unclear why the authors choose top-2 gating rather than using all four aggregators, or a different top-k setting. An ablation study comparing top-1 through top-4 would better justify this design choice.

(2) While the paper argues that using multiple aggregators helps capture morphological heterogeneity, restricting prediction to only the top 2 out of 4 experts may limit the diversity of patterns the model can leverage, potentially underusing the remaining aggregators.

(3) Although the authors claim improved interpretability, the current evidence, mainly JS divergence between attention distributions and the qualitative examples (e.g., Figure 2), does not convincingly demonstrate that the selected aggregators focus on clinically plausible or pathology-relevant regions. More direct localization-style validation would strengthen this claim.

**Detailed Comments:**

How does the performance compare to a variant that uses all aggregators (i.e., no top-k selection)?

Also, in Figure 2(C), Aggregators 1 and 2 appear to receive near-zero weight for almost all patients, while Aggregators 3 and 4 dominate the predictions. Could the authors provide more explanation for this behavior (e.g., is it expected, and does it indicate router collapse or expert redundancy)?

**Justification Of Final Rating:**

In their rebuttal, the authors added an ablation study to address my initial concerns about the choice of top-K and provided clear clarifications to my questions. Overall, I consider this a good paper and recommend it for acceptance.

**Justification Of The Preliminary Rating:**

This paper proposes the MoA framework for slide-level diagnosis in computational pathology and reports substantial performance improvements over its single-aggregator counterpart. However, the interpretability evidence is not fully convincing, and additional ablation studies would be helpful to better justify the key design choices.

**Questions To Address In The Rebuttal:**

(1) Abalation study of top K selection.
(2) Since adding more aggregators increases the parameter count, I suggest the authors report the additional training and inference overhead (e.g., parameters, memory, and runtime) introduced by the proposed approach.
(3) Please clarify the above-mentioned weakness.

---

> ### Author Response · Authors · 2026-01-25
>
> Dear Reviewer, thank you for your valuable comments and feedback. We appreciate your recognition of the paper’s strengths, particularly the motivation and comprehensive evaluation.
> Here is our point-by-point response to your questions:
>
> **Top-k decision in the router:** Thank you for pointing out the limitations of a top-2 routing. We agree with you that restricting the model to two aggregators, or even limiting the total number of aggregators to four, might reduce the diversity the model can capture in different scenarios. Following your remark, we conducted ablations for k = 1, 2, 3, and 4 (max), and included this ablation in the revised manuscript in the new Table 2. The following text is inserted in Section 4.4: “As expected, top-1 routing achieves similar performance to the baseline (78.3 vs. 78.6), while top-2 routing yields the largest improvement over the baseline (81.5). Increasing k beyond 2 does not lead to further gains.”
> Although top-2 aggregators yielded the best results with four aggregators in our ablations, the number of aggregators and the choice of top-k can be adjusted depending on the dataset and the task.
>
> **Aggregator weight distribution:** As correctly pointed out by you, the provided aggregator weight distribution for the HANCOCK dataset shows two dominant aggregators (Figure 2C). We would like to note that this example is coincidental; it is not expected that only two aggregators will always dominate. To prove this point, we have included another example of an aggregator weight distribution from the CPTAC-ALL organ classification task, which has higher heterogeneity and 10 classes. In this setting, it is more evident that different aggregators learn to specialize in different organ types which we show in the newly added Figure 5.
>
> **Explainability:** For each sample in the dataset, we calculated the attention assigned by the aggregators and the JS divergence between them (Figure 2C). The mean JS divergence across all samples is 0.42, indicating that different aggregators attend to different patterns within the same sample.
> After quantitatively demonstrating these attention differences, we qualitatively evaluated whether they are clinically relevant. We chose hematologic cytology for this purpose (i.e. the AML-Hehr dataset), since (i) pathology patches often contain complex multistructure content that can be hard to interpret, and (ii) it is hard to evaluation thousands of patches from an histology whole slide image. In contrast, the hematologic cytology dataset contains a single cell per image, with labels available for each image, and up to 500 images per patient. We present three patients with three genetic subtypes of acute myeloid leukemia with known morphological features (Figure 3).
> Following suggestions , we added another quantitative evaluation for the AML-Hehr task. We computed the intersection over union (IoU) of the top-k cells selected by different aggregators across all samples in the test set. We evaluated k as a top fraction with values 0.01, 0.05, 0.10, 0.20, 0.25, 0.50, 0.75, and 1 (approximately corresponding to the top n cells where n=5, 25, 50, 100, 125, 250, 375 and 500 cells). We included this analysis in the new Figure 4 and describe it in the text of our revised manuscript: “For quantitative evaluation, we calculated the intersection over union (IoU) between the high-attention top-k fraction of cells selected by different aggregators (Appendix Figure 4). The mean IoU across test samples is 0.16±0.17 at k=0.01 and 0.21±0.10 at k=0.05, showing that only a small fraction of highly attended cells are shared between aggregators.”
>
> **Computational overhead:** We thank the reviewer for raising this point. In Table 5, we now also report the number of parameters. Although having four times more parameters than a single aggregator, our mixture of aggregators approach requires only 2.5x more inference time (due to the top-2 strategy).

---

### Author Rebuttal · Authors · 2026-01-25

**Rebuttal:**

We sincerely appreciate the reviewers’ constructive comments and suggestions, which have greatly contributed to improving our paper. In the revised version of the manuscript, we carefully considered all feedback, presented additional experiments and incorporated suggested additions wherever possible.

In summary, we addressed the following additional suggestions: (i) ablations for top-k routing, (ii) a new quantitative analysis of attention distributions, (iii) a detailed analysis of aggregator specialization over training epochs, (iv) a comparison with simple ensembling, and (v) experiments with mixed-type aggregators and a revised related work section.

Detailed responses to each reviewer’s comments are provided separately.

We thank reviewers for their time and effort in reviewing our submission.

**Supporting Material:**

/attachment/68e87574bf71ea5d7c573a4cfd6997e36fc54ef5.pdf

---

### Meta-Review · Area_Chair_Pqqy · 2026-02-08

**Recommendation:** Accept (Poster)
**Confidence:** 4

**Metareview:**

While the methodological novelty of this work is modest, overall, the proposed approach demonstrates clear effectiveness in addressing the inherent heterogeneity of pathological images, supported by rigorous and comprehensive experimental validation.

---

### Decision · Program_Chairs · 2026-02-13

Accept (Poster)